# Study on Strength and Microstructure of Red Clay Reinforced by F1 Ionic Soil Stabilizer

Xingwei Wang [1], Jian-dong Li [1,*], Xu Wang [1,2], Yanjie Zhang [1], Daijun Jiang [1] and Guanhua Zhao [1]

[1] School of Civil Engineering, Lanzhou Jiaotong University, Lanzhou 730070, China
[2] National and Provincial Joint Engineering Laboratory of Road & Bridge Disaster Prevention and Control, Lanzhou 730070, China
* Correspondence: jdli620523@163.com

**Abstract:** High-liquid limit red clay has poor engineering characteristics, namely poor water stability, low strength, and large expansion and contraction deformation. The clay may be reinforced with an F1 ionic soil stabilizer. The engineering characteristics of this reinforced clay were studied, specifically concerning its basic physical parameters, shear strength parameters, and micropore structure. The F1 ionic soil stabilizer significantly improved the water sensitivity, compaction characteristics, and shear strength of red clay. We determined that the optimal F1 ionic soil stabilizer mix was $0.5 \, \text{L/m}^3$, resulting in a reinforced clay with plastic limit increased by 45.74%, optimal moisture content increased by 12.12%, maximum dry density increased by 5.8%, liquid limit reduced by 8.4%, plasticity index reduced by 43.8%, infiltration coefficient reduced by 41.8%, cohesion increased 1.64-fold, and internal friction angle increased 1.30-fold. Freeze-thaw cycles reduced the shear strength parameters of the reinforced red clay, although even after 15 cycles, it still had 18.4% higher cohesion and 57.1% higher internal friction angle than undisturbed red clay. The F1 ionic soil stabilizer significantly reduced the pore size and area of red clay, the complex connected pore structure is adjusted to a more regular structure. The reinforced clay had 56.64% lower pore area ratio, 32.27% lower average Feret diameter, and 2.43% lower fractal dimension.

**Keywords:** red clay; ionic soil stabilizer; physical and mechanical properties; microstructure

## 1. Introduction

Red clay is a carbonate weathering product rich in strongly hydrophilic clay minerals such as montmorillonite, illite, and kaolinite [1]. It has several undesirable engineering properties such as swelling and softening in water, shrinkage and dry cracking upon drying, poor water stability, low strength, and large deformation [2–4]. When directly used as an engineering filler in structures, it often causes issues such as uneven foundation settling, subgrade collapse, and slope instability [5,6]. Red clay could threaten the service performance and life cycle of highways, railways, reservoirs, and other engineering structures in areas with a red clay subsurface; therefore, research into red clay reinforcement and augmentation is very important to engineers.

Lime, cement, and other traditional binding materials are widely used in red clay reinforcement because of their low cost, abundance, and availability in bulk quantities. Amoudi, Liu, and Hu studied the physical and mechanical properties of lime-modified red clay and found that traditional cementitious materials can significantly improve the physical properties and strength of red clay [7–9]. Tan studied the influence of particle size on the bearing ratio of lime-modified red clay and found that its strength and water stability could be significantly improved given careful control of its particle size [10]. Alhassan, Liu, and Xiao stabilized red clay with cement and found that a reasonable amount of cement could significantly improve its hydraulic properties and mechanical strength [11–13].

While conventional binding agents are proven to work, they come with a high carbon footprint and emission. As an alternative, Yang used sand and gravel to improve the special

engineering properties of red clay [14]. Yang found that a composite curing agent can significantly improve red clay's water stability and bearing capacity [15]. Chen studied the mechanism of red clay reinforcement by calcium carbonate nanoparticles from the perspective of the colloidal chemistry of mineral particles [16,17]. Liu studied the mechanical properties and microstructure of modified red clay at different pH levels and found that acidity and alkalinity can significantly affect the degree of intergranular interaction in red clay [18]. Yan augmented red clay with rubber particles from waste tires and increased shear strength [19].

Ionic soil stabilizers are a new industrial product that require a lower dosage, cost less, reinforce the soil well, and work across various soil varieties. Cui, Xiang, and You found that ionic soil stabilizers can reduce the plasticity index and improve the compactness and strength of red clay [20–22]. Marto found that SS299 soil stabilizer can significantly improve the strength of red clay [23]. Liu and Zhao found that ionic soil stabilizers can change the type and concentration of ions in soil with excess pore water, thereby reducing the thickness of water film and disrupting the electrical double layer through a series of physicochemical reactions including ion exchange, adsorption, and encapsulation [24,25]. Katz found that the reinforcing effect of ionic soil stabilizers is influenced by the soil's mineral composition [26]. Li used F1 ionic soil stabilizer to reinforce loess and found that F1 can reduce the thickness of water film and significantly increase the dry density and compressive strength of loess [27,28].

Since the 20th century, researchers have intensively studied the reinforcement of expansive soil, loess, and other special soils with ionic soil stabilizers. Despite the long history, the research effort has not left the experimental exploration stage, and the concept has not seen wide commercial adoption. Therefore, studying the engineering characteristics and strengthening mechanisms of red clay reinforcement is a top priority. In this study, sulfonated acrylic polymer formula F1 ionic soil stabilizer (F1 for short) was used to strengthen Gansu red clay, and its physical and mechanical properties with different dosages of F1 were analyzed. Its microscopic morphology was inspected via SEM. The data gathered here is expected to pave the way for the wider adoption of ionic soil stabilizers in regions with red clay substratum.

## 2. Materials and Experiment Program

### 2.1. Materials

Red clay was extracted from the subgrade fill of a highway in Gansu Province. The soil depth was less than 10 m, and the core sample was brick red to reddish brown. The sample was sifted through a 2 mm sieve, revealing that the clay contained fine particles of about 89.21%, had argillaceous cementation, weak expansibility, and low liquid limit. The basic physical parameters are displayed in Table 1.

**Table 1.** Basic physical properties of red clay.

| Plasticity Limit (%) | Liquid Limit (%) | Plasticity Index (%) | Permeability Coefficient (cm·s⁻¹) | Optimum Moisture Content (%) | Maximum Dry Density (g·cm⁻³) | Particle Size Distribution (%) | | |
|---|---|---|---|---|---|---|---|---|
| | | | | | | Sand Size (mm) 2~0.075 | Silt Size (mm) 0.075–0.005 | Clay Size (mm) <0.005 |
| 18.8 | 43.8 | 25 | $9.2702 \times 10^{-9}$ | 19.76 | 1.71 | 10.79 | 51.71 | 37.5 |

### F1 Ionic Soil Stabilizer

At room temperature, the F1 ionic soil stabilizer is a viscous light-yellow liquid with a density of 1.35 g/cm³. The main active component is acrylic sulfonated organic polymer accompanied by a variety of acidic surfactants. When F1 is used to reinforce soil, it first needs to be diluted; the dilution ratio recommended by the manufacturer is 1:200~1:300. When F1 comes into contact with water, it can rapidly dissociate, releasing high-energy hydrogen ions ($H_3O^+$) with a small hydration radius, high osmotic pressure, and strong potential. According to the cation exchange rules (exchange potential energy: $Na^+ < Li^+ < K^+ < Mg^{2+} < Ca^{2+} < Ba^{2+} < Cu^{2+} < Al^{3+} < Fe^{3+} < H_3O^+$) the ion exchange re-

action between metal cations and polar water molecules with larger hydration radius on clay particles would occur, reducing the thickness of bound water film and shrinking interparticle gap [27,28]. Soil particles floc and agglomerate, lowering the soil's water sensitivity and improving compactness, load bearing capacity, water stability, and engineering characteristics [25–28]. Figure 1 illustrates the reinforcement diagram.

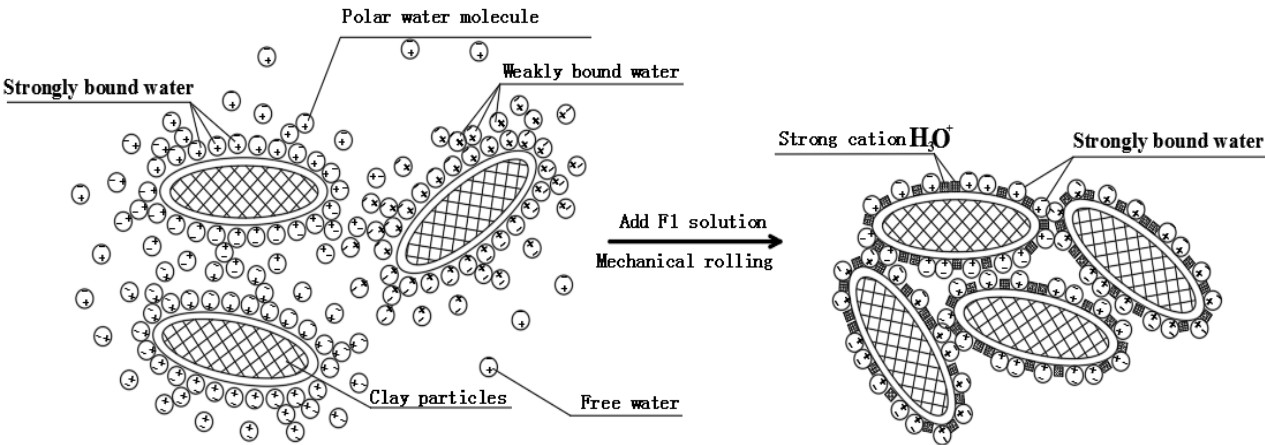

**Figure 1.** Microscopic schematic diagram of F1 reinforced soil.

### 2.2. Specimen Preparation and Testing Program

### 2.2.1. Physical Parameter Test

The red clay sample was air-dried, then sifted through a 2 mm sieve. A dose of F1 was prepared according to the soil mass required for the test and the desired mixing ratio, then diluted with water at a 1:200 volume ratio and mixed evenly into the red clay sample. Reinforced red clay samples were prepared containing 0.0, 0.3, 0.5, and 0.7 L/m³ of F1 according to the specification for highway geotechnical test [29]. The reinforced clay samples were put through the liquid plastic limit joint determination, compaction, and expansion tests.

### 2.2.2. Unconsolidated Undrained Triaxial Test (UU)

Following the compaction test results, the stabilized red clay samples with different F1 doses were humidified to the optimal moisture content and stored in sealed bags for 24 h to allow the moisture to distribute uniformly. To prepare the triaxial specimens, the soil samples were packed into cylindrical steel molds and pressed to 95% compactness using a hydraulic press in three layers, obtaining cylindrical samples 39.1 mm in diameter and 80 mm in height, as illustrated in Figure 2. There are 3 samples in each group, totaling 48 samples.

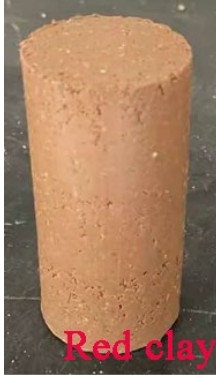 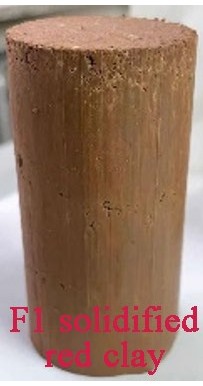

**Figure 2.** Triaxial specimen.

### 2.2.3. Freeze-Thaw Cycle Test

The freeze-thaw cycle test was performed using the same triaxial specimens described previously, with three specimens in each group for 60 specimens. The specimens were enclosed in a sealed bag and labeled with a number. They were subjected to 12 h freezing at −20 °C and 12 h thawing at 20 °C in a DW-40 low-temperature test chamber. Three specimens from each F1 dose group were taken to the triaxial UU test at each time point: at the start of the experiment (cycle 0) and at cycle 1, 5, 10, and 15. The shear rate was set at 0.8 mm/min, and the confining pressure was 50, 100, 200, and 400 kPa. The triaxial UU test was performed using the triaxial apparatus demonstrated in Figure 3 to calculate the shear strength parameters of the F1 reinforced red clay with different doses, a different number of freeze-thaw cycles, and the influence of the number of freeze-thaw cycles on the mechanical strength of F1 reinforced red clay was analyzed.

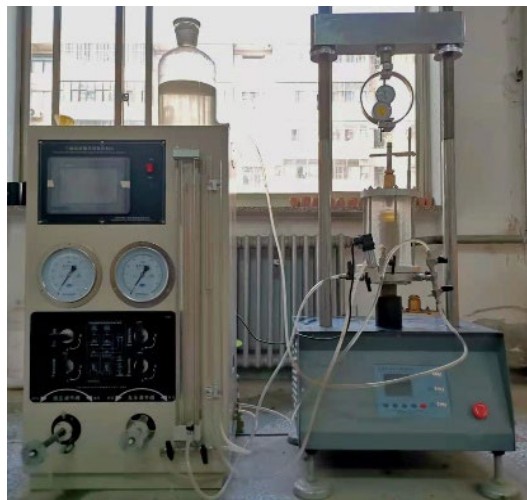

**Figure 3.** Strain-controlled triaxial apparatus.

### 2.3. Scanning Electron Microscope

A small sample from each F1 dose group was taken and air-dried at room temperature. The JSM-5600LV electron microscope displayed in Figure 4 was used to obtain 1000× magnification SEM images of the F1 red clay samples. Image Pro Plus (IPP) microstructure analysis software was used to analyze the micropore morphology of F1 red clay and to compare it with virgin red clay [26–28].

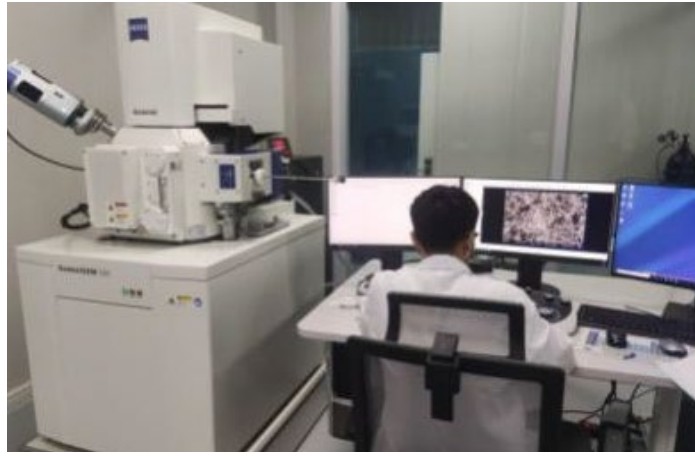

**Figure 4.** JSM-5600LV electron microscope.

## 3. Results and Discussion

### 3.1. Effect of F1 on Basic Physical Properties of Red Clay

The basic physical parameters of solidified red clay with different F1 doses are displayed in Table 2. As the F1 dose increased, the clay's plastic limit increased initially and then decreased. The liquid limit and plasticity index decreased at first and then increased. The samples with 0.5 L/m$^3$ F1 exhibited a 45.74% higher plastic limit, 8.4% lower liquid limit, and 43.8% plasticity index than the virgin red clay, indicating the clay's water adsorption capacity and water sensitivity was reduced by F1. The F1 can hydrolyze to produce a high concentration of $H_3O^+$ with a strong exchange potential energy, which can disrupt the electric double layer through ion exchange. The result was a thinner adsorbed water film and smaller inter-particle distance, which modified the soil's micropore structure and the interaction between soil grains, weakened capillary action, and resulted in increased plastic limit and reduced liquid limit and plasticity index [27,28].

**Table 2.** Physical parameters of solidified red clay with different F1 content.

| F1 Dosage/L·m$^{-3}$ | Plastic Limit/% | Liquid Limit/% | Plasticity Index | Optimum Moisture Content/% | Maximum Dry Density/g·m$^{-3}$ | Permeability Coefficient/cm·s$^{-3}$ |
|---|---|---|---|---|---|---|
| 0 | 18.8 | 43.8 | 22.6 | 19.76 | 1.71 | $15.72 \times 10^{-9}$ |
| 0.3 | 24.0 | 41.4 | 19.8 | 21.34 | 1.74 | $9.32 \times 10^{-9}$ |
| 0.5 | 27.4 | 40.1 | 12.7 | 22.18 | 1.81 | $9.15 \times 10^{-9}$ |
| 0.7 | 24.8 | 41.3 | 16.4 | 20.67 | 1.76 | $9.56 \times 10^{-9}$ |

A similar trend could be observed in optimal water content, maximum dry density, and permeability coefficient, which also exhibited extrema around an F1 dosage of 0.5 L/m$^3$. At this dose, the optimum moisture content was 12.12% higher, the maximum dry density was 5.8% higher, and the permeability coefficient was 41.8% lower than virgin red clay. These effects were because F1 had reduced the thickness of combined water film and inter-particle spacing through ion exchange and the hydrophobic action of sulfonated oil, enhancing particle agglomeration under compaction, increasing compactnessm and reducing the permeability coefficient. The reduced water film also reduced water sensitivity, increased the resistance between particles, and increased the optimal moisture content of solidified red clay.

### 3.2. Unconsolidated Undrained Triaxial Compression Test Results

The shear strength parameter is an important mechanical index of F1 reinforced red clay. As demonstrated in Figure 5, in combination with the UU test results, the $(\delta_1-\delta_3)_{f}-\varepsilon$ curve was drawn for red clay samples with different F1 doses. When the confining pressures were 50, 100, 200, and 400 kPa, respectively, the deviator stress of red clay with different F1 contents increased with increasing axial strain, and the relationship between them was approximately hyperbolic. According to the specification [29], when there is no obvious peak, the principal stress difference corresponding to 15% axial strain is taken as the failure stress.

The relationship between the maximum deviator stress of F1 reinforced red clay and F1 content under various confining pressures is displayed in Figure 6. The maximum deviator stress of F1 reinforced red clay under different confining pressures increased at first and then decreased in response to F1 dose. Clay samples with 0.3 L/m$^3$ F1 exhibited 1.13-fold, 1.41-fold, 1.50-fold, and 1.12-fold higher maximum deviator stress at 50, 100, 200, and 400 kPa confining pressures, respectively, compared to virgin red clay. Under the same parameters, clay samples with 0.5 L/m$^3$ F1 exhibited 1.67-fold, 1.69-fold, 1.97-fold, and 1.46-fold higher maximum deviator stress. At an F1 dose greater than 0.5 L/m$^3$, the maximum deviator stress decreased. According to the analysis results in Table 2, 0.5 L/m$^3$ was the best F1 dose for reinforced red clay.

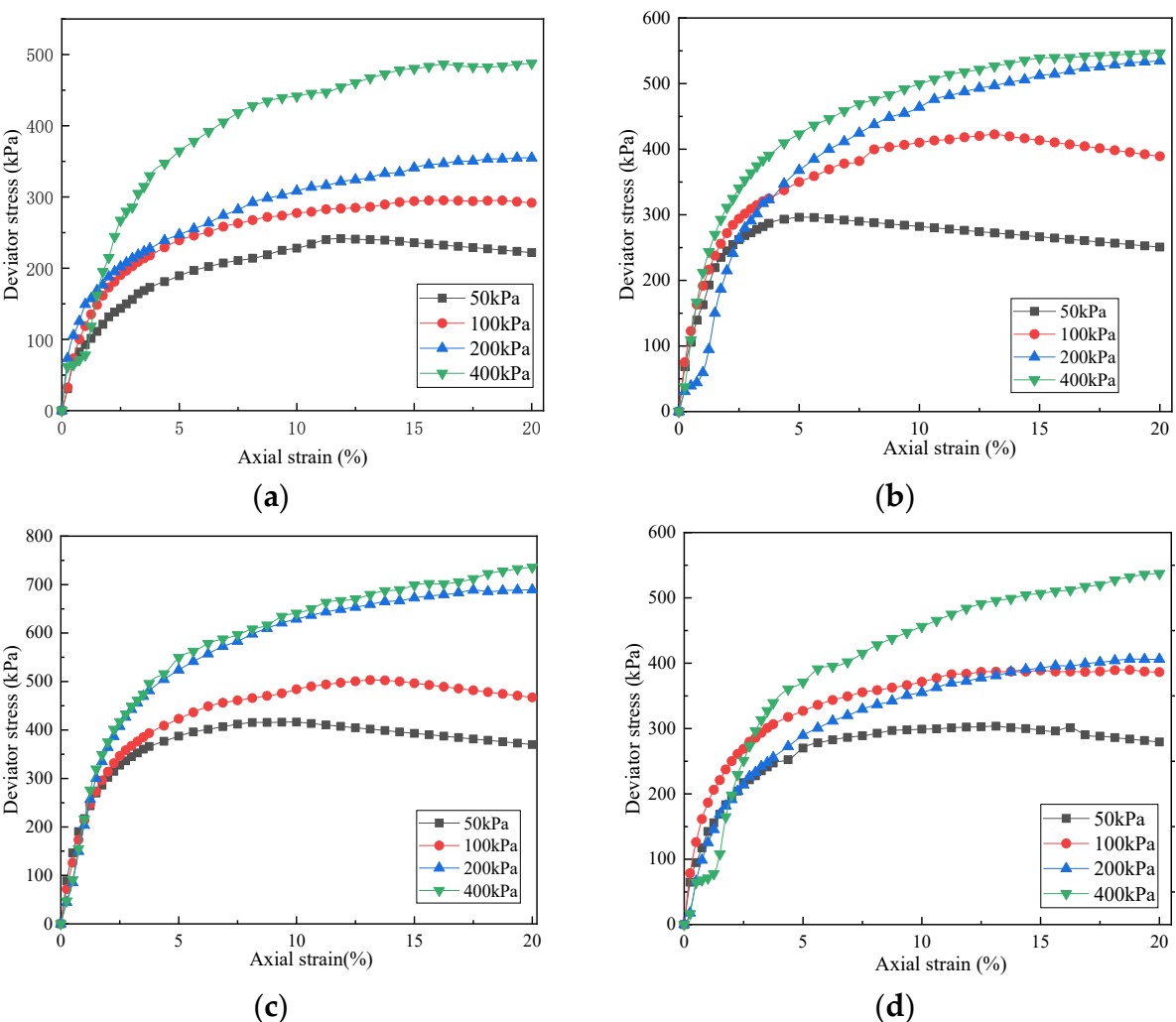

**Figure 5.** Stress-strain curves of red clay with different dosages of F1 agent. (**b**) 0.0 L/m³. (**b**) 0.3 L/m³. (**c**) 0.5 L/m³. (**d**) 0.7 L/m³.

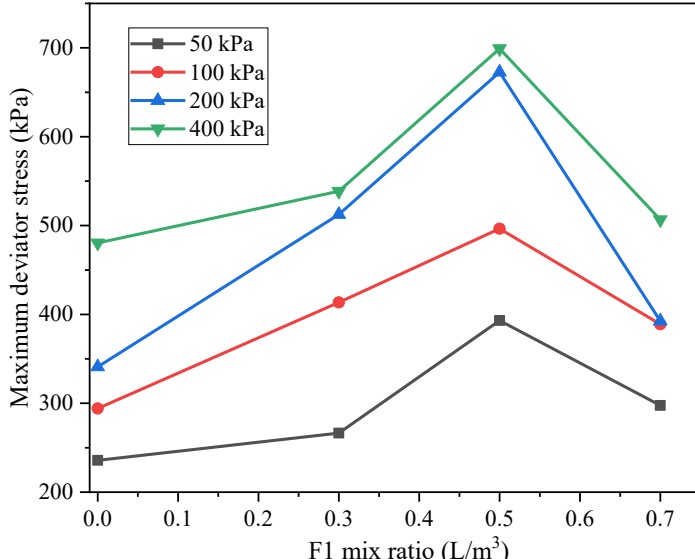

**Figure 6.** Relationship between maximum deviator stress of red clay and F1 content.

The molar stress circle of F1 reinforced loess sample under different confining pressures at the time of failure was drawn, and the cohesion $c$ and internal friction angle $\varphi$ of reinforced red clay with different F1 doses were obtained as illustrated in Figure 7. The cohesion and internal friction angle of red clay initially increased and then decreased alongside the F1 dose. The 0.3 L/m³ F1 clay exhibited 1.32-fold higher cohesion and 1.12-fold higher internal friction angle, while the 0.5 L/m³ F1 clay exhibited 1.64-fold higher cohesion and 1.30-fold higher internal friction angle. The increase in cohesion and internal friction angle was because F1 can effectively reduce particle spacing and water film thickness, thereby increasing cohesion and soil compactness and, in turn, increasing the soil's shear strength parameters.

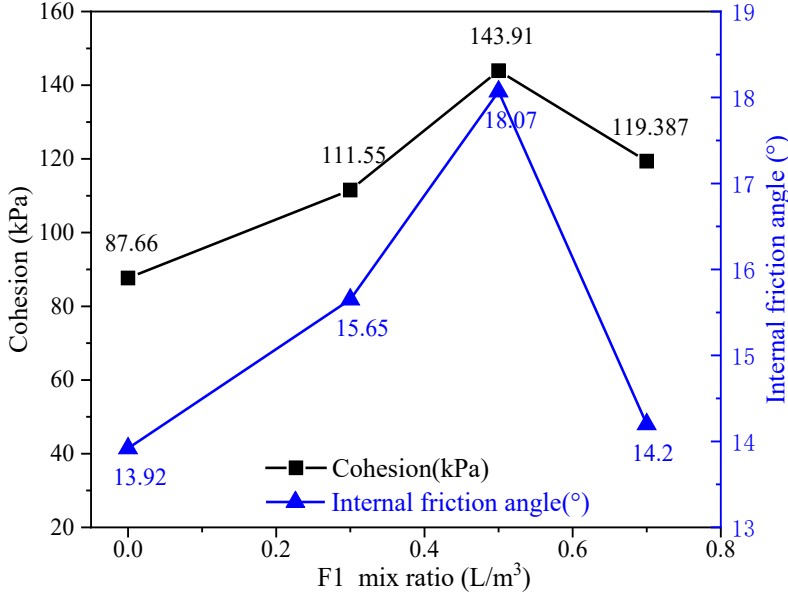

**Figure 7.** Relationship between shear strength parameters of red clay and F1 content.

### 3.3. Effect of the Freeze-Thaw Cycle on Shear Strength Parameters of F1 Reinforced Red Clay

Freeze-thaw cycles can compromise the structure of compacted soil, reducing its mechanical strength, stability, and load-bearing capacity [30]. To analyze the effect of freeze-thaw cycles on the strength of F1 reinforced red clay, the relationship curve between the shear strength and the number of freeze-thaw cycles under different dosages was drawn, as displayed in Figure 8.

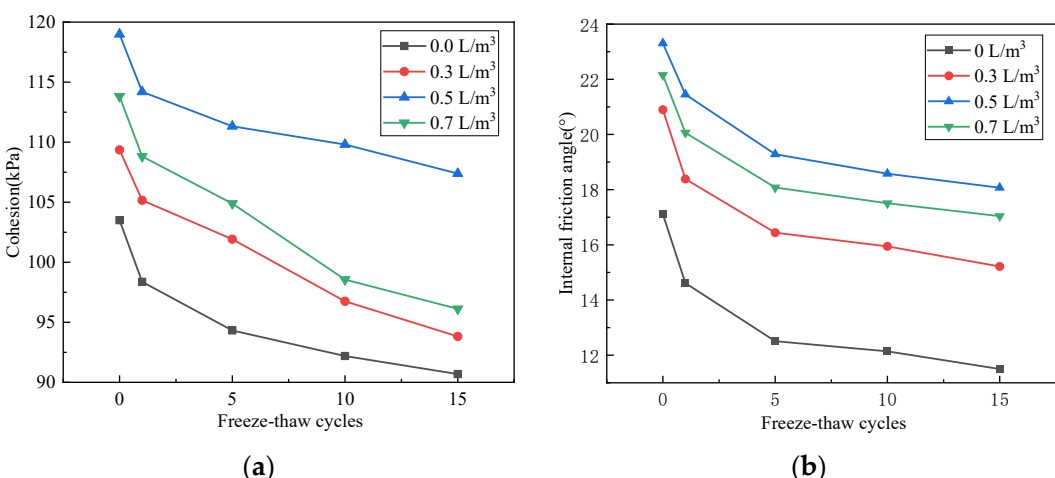

**Figure 8.** Relationship between shear strength and number of freeze-thaw cycles. (**a**) Cohesion. (**b**) Internal friction angle.

Figure 8 shows that the F1 red clay's shear strength decreased as the sample went through successive freeze-thaw cycles. After 15 freeze-thaw cycles, the 0.3, 0.5, and 0.7 L/m$^3$ F1 clay samples still exhibited 3.5%, 18.4%, and 6.0% higher cohesion than the virgin red clay, respectively, and correspondingly 32.3%, 57.1%, and 48.2% higher internal friction angle, respectively. These measurements indicated that the F1 reinforcing effect survived at least 15 freeze-thaw cycles and that the 0.5 L/m$^3$ F1 doses provided the best reinforcement effect. The freeze-thaw cycles caused a corresponding expansion and contraction, which disrupted the intergranular interaction in the red clay and weakened the cementation and biting force between particles, resulting in lower cohesion and internal friction angle after each successive cycle. The F1 soil stabilizer can promote the flocculation and agglomeration of soil particles through ion exchange and hydrophobic action, increasing the bonding strength between particles, reducing water sensitivity, and mitigating the damaging effect of the water phase change in red clay.

## 4. Microstructure Analysis of F1 Reinforced Red Clay

Figure 9 displays the SEM images of red clay with different F1 dosages processed using the IPP micro image analysis software, where red represents pores, and gray represents soil skeleton.

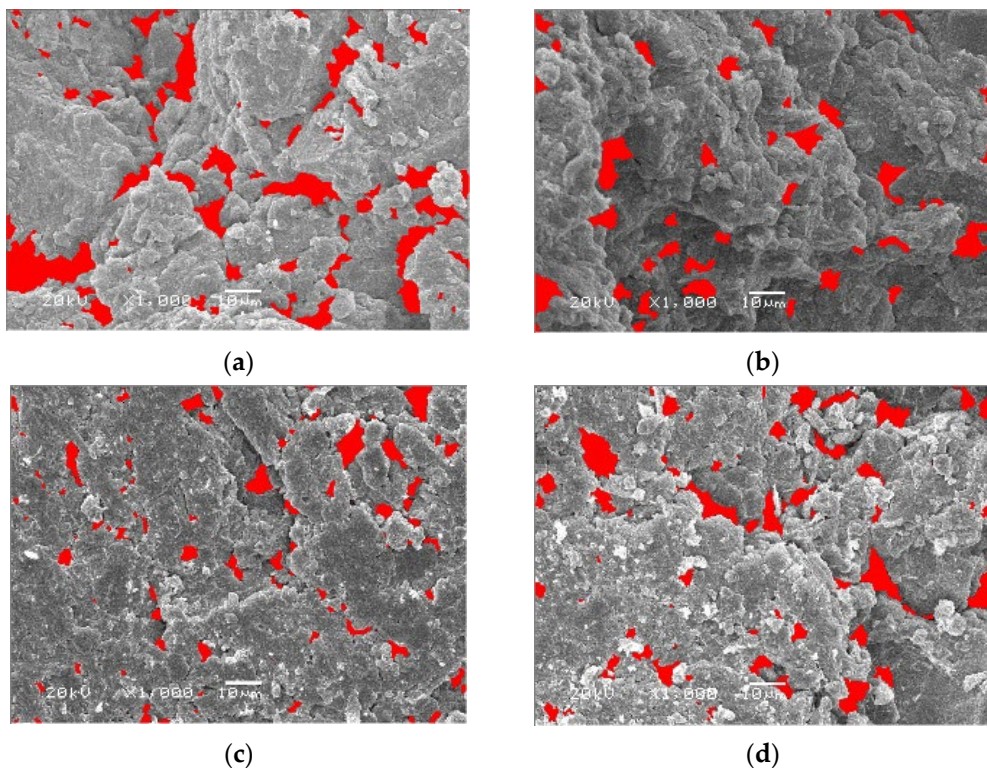

**Figure 9.** Microscopic pore distribution of F1 reinforced red clay (1000 times). (**a**) Undisturbed red clay. (**b**) 0.3 L/m$^3$. (**c**) 0.5 L/m$^3$. (**d**) 0.7 L/m$^3$.

The soil particles in virgin red clay were randomly distributed in flakes or agglomerates, and the pore structure was well developed, most of which were large and medium pores with good connectivity. In Figure 9b,c, adding F1 increased aggregation and significantly reduced the number and size of large and medium pores. The inter-particle connection was mainly in compact stacking and agglomeration, and the degree of inter-particle cementation and the compactness of soil mass were significantly increased. In Figure 9d, at 0.7 L/m$^3$ of F1, the agglomeration and cementation of soil particles were weakened, and the aggregates were reduced and distributed in a fragmented manner. The reason was that the excess strong cationic charge generated charge repulsion forces between the cations on the surface of the soil particles, increasing the spacing of soil particles and

weakening the degree of cementation between particles. The result depicted an optimal dosage when F1 is used to reinforce red clay.

To quantitatively analyze the complexity of the micropore structure of red clay before and after F1 reinforcement, the Image J software was used to extract the micropore structure parameters of F1 reinforced red clay and draw the change regulation of pore area ratio, average Feret diameter, and fractal dimension, as displayed in Figure 10.

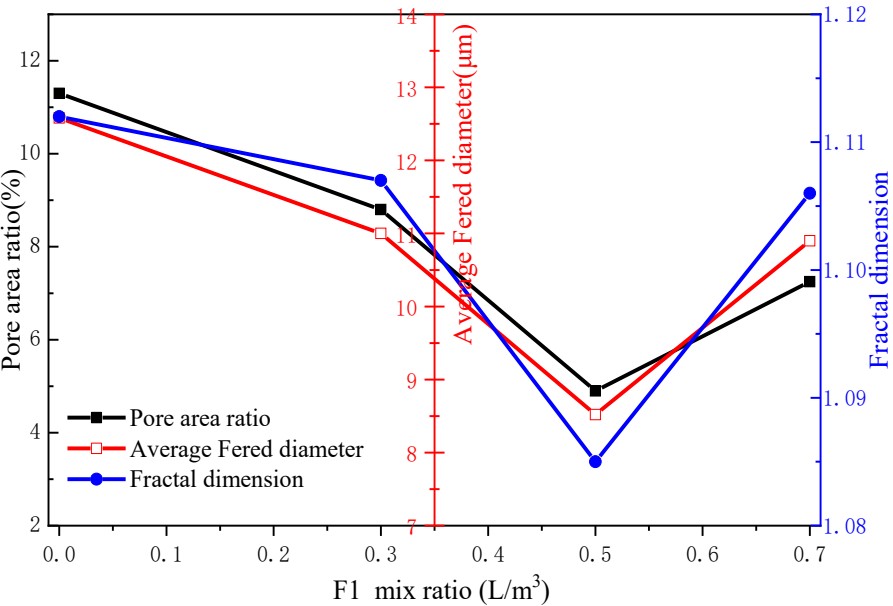

**Figure 10.** Relationship curve between micropore structure parameters of red clay and F1 content.

The pore area ratio, average Feret diameter, and fractal dimension of red clay exhibited a change regulation of first decreasing and then increasing with increased F1 content. The 0.5 L/m$^3$ F1 sample had a 56.64% lower pore area ratio, 32.37% lower average Feret diameter, and 2.43% lower fractal dimension compared to the virgin red clay, indicating that the pore size and area of F1 reinforced red clay were significantly reduced and that the pore morphology tended to be regular from the complex connected band structure. The reason was that the F1 reinforcement changed the contact mode between the red clay particles, enhancing the cementation characteristics between particles, significantly reducing the micropore area and connectivity, and improving the physical and mechanical properties of red clay [23,26].

## 5. Discussion

In this study, the results show that at an F1 dose of 0.5 L/m$^3$, the plastic limit, optimal water content, and maximum dry density increased by 45.74%, 12.12%, and 5.8%, respectively, and the liquid limit, plasticity index, and permeability coefficient decreased by 8.4%, 43.8%, and 41.8%, respectively, and the cohesion and internal friction angle increased by 1.64 times and 1.30 times, compared to virgin red clay.

Related studies on ionic soil stabilizer show that F1 can significantly improve the water sensitivity, compaction properties, and micropore structure of red clay through ion exchange, destruction of double electric layer structure and hydrophobic interaction, and improve the compactness, strength, and freezing resistance of solidified soil.

F1 has many advantages that traditional soil stabilizers do not have, so it has a broad application prospect in soil reinforcement engineering. Related studies have found that durability is an important disadvantage of ionic soil stabilizers. In this paper, the physical and mechanical properties of F1-reinforced red clay were studied only through laboratory tests. In the future, the influence of wetting and drying cycles and the engineering characteristics of F1-reinforced red clay need to be studied.

## 6. Conclusions

The basic physical parameters, shear strength parameters, and the change regulation of the micropore structure of red clay reinforced with F1 ionic soil stabilizer were studied through the basic physical parameter test UU test, freeze-thaw cycle test, and SEM image analysis. The following conclusions were drawn.

(1) With the increase of F1 content, the plastic limit, permeability coefficient, optimum moisture content, and maximum dry density of solidified red clay first increased and then decreased, and the liquid limit and plasticity index first decreased and then increased, indicating that F1 can significantly improve the water sensitivity and compaction characteristics of red clay.

(2) With the increase of F1 content, the maximum deviator stress, cohesion, and internal friction angle of F1 reinforced red clay show a change regulation of first increasing and then decreasing, and $0.5 \text{ L/m}^3$ was the best F1 dose. Compared with undisturbed red clay, the cohesion and internal friction angle of the red clay strengthened with the optimal dosage increased by 1.64 times and 1.30 times, respectively, indicating that F1 can significantly increase the shear strength of the red clay.

(3) The shear strength parameters of F1 reinforced red clay decreased with successive freeze-thaw cycles. Compared with undisturbed red clay, the cohesion and internal friction angle of solidified red clay were 18.4% and 57.1% higher, respectively, with the optimal dosage of F1 $0.5 \text{ L/m}^3$ after 15 freeze-thaw cycles, indicating that F1 can still significantly increase the strength of red clay after 15 freeze-thaw cycles.

(4) The pore area ratio, average Feret diameter, and fractal dimension of red clay exhibited a change regulation of first decreasing and then increasing with increased F1 content. Compared to undisturbed red clay, the pore area ratio, average Feret diameter, and fractal dimension of the reinforced red clay with the optimal F1 content of $0.5 \text{ L/m}^3$ were reduced by 56.64%, 32.27%, and 2.43%, respectively, indicating that the pore size and area of the red clay after F1 reinforcement were significantly reduced, and the pore morphology tended to be regular.

**Author Contributions:** Conceptualization, X.W. (Xingwei Wang) and J.-d.L.; methodology, J.-d.L.; software, X.W. (Xingwei Wang); validation, J.-d.L., Y.Z. and G.Z.; formal analysis, X.W. (Xingwei Wang); investigation, D.J.; resources, X.W. (Xu Wang); data curation, J.-d.L.; writing—original draft preparation, X.W. (Xingwei Wang); writing—review and editing, X.W. (Xu Wang) and J.-d.L.; supervision, X.W. (Xu Wang); project administration, X.W. (Xu Wang); funding acquisition, Y.Z. All authors have read and agreed to the published version of the manuscript.

**Funding:** This research was funded by the National Natural Science Foundation of China grant number 51868038 and 52268058, project supported by the Young Scholars Science Foundation of Lanzhou Jiaotong University grant number 2022054 and Gansu Province Youth Science and Technology Fund Program grant number 22JR5RA369.

**Institutional Review Board Statement:** This study did not involve humans or animals.

**Informed Consent Statement:** This study did not involve humans.

**Data Availability Statement:** This study did not report any data.

**Conflicts of Interest:** The authors declare no conflict of interest.

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
