# Peer review of "Study on Strength and Microstructure of Red Clay Reinforced by F1 Ionic Soil Stabilizer"

_applsci, doi:10.3390/app12199831_

Round 1

Reviewer 1 Report

The paper presents an experimental study on the impact of F1 ionic soil stabilizer on the water sensitivity, compaction characteristics, and shear strength of red clay. The paper first provides the materials and experimental setup. Then it discuss about various impacts of F1.

While there are several places in this paper that can be improved, the overall quality and completeness of the paper are excellent. Therefore, the manuscript is recommended to publish after minor revision.

Here are some questions and comments for the paper:

1. In the first page, there are multiple highlighted texts with yellow color. It's not clear why they're highlighted.

2. In abstract, this sentence "..., modifying its pore morphology from the complex connected band structure to be more regular" is confusing

3. In introduction, more details need to be given for "Amoudi, Liu and Hu studied the physical and mechanical properties of lime-modified red clay [7-9]."

4. In introduction, the paper says that conventional binding agents have the disadvantage of "high carbon footprint and emission", but it doesn't explain how ionic soil stabilizers do not have such disadvantage

5. In introduction, this sentence "..., that require a lower dosage, cost less, reinforce the soil well, and work across various soil varieties", should be "requires", "costs", "reinforces" and "works"

6. In discussion and conclusion, the sentence "the pore morphology tended to be regular from the complex connected band structure" is confusing. "regular" is not a antonym of "complex"

Reviewer 2 Report

The article investigates the effect of F1 ionic soil stabilizer on the engineering characteristics of red clay. The optimal content of this additive was determined, at which the best frost resistance and maximum strength properties are provided. Research methods are described in detail. Results are well presented and they are of interest to readers. 

However, the article needs some revision. The discussion should be moved to a separate section. It should indicate how the results are consistent with similar studies by other authors. In particular, in the introduction it was indicated that the F1 additive was previously used to modify the properties of loess soils. 

It would not be superfluous to compare the effectiveness of the F1 additive with other methods for improving the properties of red clay. Also in the discussion section, prospects for further research can be outlined.
